# Improved Mechanical Properties and Bioactivity of Silicate Based Bioceramics Reinforced Poly(ether-ether-ketone) Nanocomposites for Prosthetic Dental Implantology

**DOI:** 10.3390/polym14081632

**Published:** 2022-04-18

**Authors:** Noha Taymour, Amal E. Fahmy, Mohamed Abdel Hady Gepreel, Sherif Kandil, Ahmed Abd El-Fattah

**Affiliations:** 1Department of Substitutive Dental Sciences, College of Dentistry, Imam Abdulrahman Bin Faisal University, Dammam 31441, Saudi Arabia; ntyoussef@iau.edu.sa; 2Department of Dental Materials, Faculty of Dentistry, Alexandria University, Azarita, Alexandria 21526, Egypt; amal_ezzeldin99@yahoo.com; 3Department of Materials Science and Engineering, Egypt-Japan University of Science and Technology (E-JUST), New Borg El-Arab City 21934, Egypt; geprell@yahoo.com; 4Department of Materials Science, Institute of Graduate Studies and Research, Alexandria University, El-Shatby, Alexandria 21526, Egypt; s.kandil@usa.net; 5Department of Chemistry, College of Science, University of Bahrain, Sakhir P.O. Box 32038, Bahrain

**Keywords:** PEEK, bioglass, forsterite, nanocomposites, elastic modulus, microhardness, flexural strength, bioactivity, dental implant

## Abstract

Polyether-ether-ketone (PEEK) biomaterial has been increasingly employed for orthopedic, trauma, spinal, and dental implants due to its biocompatibility and in vivo stability. However, a lack of bioactivity and binding ability to natural bone tissue has significantly limited PEEK for many challenging dental implant applications. In this work, nanocomposites based on PEEK reinforced with bioactive silicate-based bioceramics (forsterite or bioglass) as nanofillers were prepared using high energy ball milling followed by melt blending and compression molding. The influence of nanofillers type and content (10, 20 and 30 wt.%) on the crystalline structure, morphology, surface roughness, hydrophilicity, microhardness, elastic compression modulus, and flexural strength of the nanocomposites was investigated. The scanning electron microscopy images of the nanocomposites with low nanofillers content showed a homogenous surface with uniform dispersion within the PEEK matrix with no agglomerates. All nanocomposites showed an increased surface roughness compared to pristine PEEK. It was found that the incorporation of 20 wt.% forsterite was the most effective in the nanocomposite formulation compared with bioglass-based nanocomposites; it has significantly improved the elastic modulus, flexural strength, and microhardness. In vitro bioactivity evaluation, which used biomimetic simulated body fluid indicated the ability of PEEK nanocomposites loaded with forsterite or bioglass nanofillers to precipitate calcium and phosphate bone minerals on its surface. These nanocomposites are expected to be used in long-term load-bearing implant applications and could be recommended as a promising alternative to titanium and zirconia when used as a dental implant material.

## 1. Introduction

Missing teeth with their supporting oral tissues are considered one of the major concerns in modern dentistry. Substituting single missing teeth, particularly in the anterior region, has always been an encounter for dentists [1]. With increasing patient demands, removable partial dentures have become unsatisfactory and various patients now oppose the preparation of intact teeth for the fabrication of a fixed partial denture [2]. Dental materials offer a reliable alternative that improves the quality of life for most patients with tooth loss. Among various dental materials and their efficacious applications, a dental implant is a good example of the integrated system of science and technology involved in multiple disciplines, including surface chemistry, surface topography, and biomechanics [3,4].

Traditionally, metals or ceramics are the most prevalent dental materials for use in dentistry. In particular, titanium (Ti) and its alloys were typically used over the past decades as orthopedic/dental implants due to their low density, high mechanical strength, excellent corrosion resistance, and cytocompatibility [5,6]. Nevertheless, there are concerns regarding the liberation of harmful metal ions and radiopacity of metal alloys in vivo [6,7]. Furthermore, the elastic moduli of metal alloy divergence in mechanical properties between metals and human bones causes bone resorption [8,9]. Indeed, serious post-operative complications, for instance, allergenicity, osteolysis, and loosening as well as eventual implant failure may take place [10,11]. To overcome these limitations and minimize adverse post-implantation biological reactions, substitutes for metals are broadly pursued [12,13,14].

Polymeric materials have been widely used for various applications in clinical dentistry. They offer a unique combination of properties and potentials that address fundamentally important issues of dental implants. Polyether-ether-ketone (PEEK) and polyether-ketone-ketone (PEKK) are both parts of the polyaryl-ether-ketone (PAEK) family of ultra-high performance thermoplastic polymers. PEEK is a semi-crystalline organic polymer exhibiting a highly stable chemical structure, which has been progressively employed for orthopedic and dental implants [15,16]. From an engineering perspective, PEEK possesses superior properties, such as stiffness, strength, toughness, thermal stability, resistance to chemical and radiation damage as well as non-cytotoxicity and in vivo stability, which make it a multipurpose biomaterial significantly convenient for the development of medical device application [17,18]. Moreover, PEEK has a low elastic modulus (3–4 GPa) compared to Ti and other metal alloys, which decreases the degree of stress shielding that is frequently observed in Ti-based metallic implants [19,20]. It is worth mentioning, that the elastic modulus of PEEK could be modified by adding reinforcing nanofillers to achieve an elastic modulus of 18 GPa, similar to that of human bone [21,22]. From the processing standpoint, PEEK can be readily fabricated by conventional plastic processing techniques to fit the shape of bones and is feasibly sterilized using γ-radiation, ethylene oxide gas, and steam without altering its mechanical characteristics or biocompatibility [23,24].

Although PEEK implants have been used frequently in modern medical operations, PEEK itself is chemically hydrophobic and biologically inert [25,26,27]. Subsequently, its direct binding and complete integration with the neighboring bone after implantation is very limited [28]. The bioactivity and mechanical performance of PEEK can be improved by incorporating bioactive inorganic nanoparticles to produce PEEK-based nanocomposites [29,30]. Many researchers have effectively fabricated PEEK nanocomposites using hydroxyapatite (Ca_10_(PO_4_)_6_(OH)_2_, HA) because HA is a constituent of living bone [31]. PEEK/HA nanocomposites have revealed enhanced cell and tissue responses compared to pristine PEEK in previous studies. However, critical implant failures often occur at the implant-abutment interface [22].

Silicate-based bioceramics, such as bioglass (BG), forsterite (FT), calcium silicate, bredigite, and akermanite have received increasing attention to treat bone defects. BG has excellent bioactivity and biocompatibility. It is used in several biomedical applications, for example, non-load-bearing implants, bone cement, tissue engineering scaffolds, drug delivery systems, and bioactive coatings [31,32]. In particular, composite materials incorporating BG are being considered promising for biomedical applications. By modifying the composition, BG can be designed to degrade at a controlled rate that matches the development of new bone tissue, without any toxic effects. Moreover, BG is capable of turning on the body’s own regenerative system to stimulate bone growth [33].

FT is a new bioceramic that has demonstrated good biocompatibility. Researches show that FT also possesses mechanical properties superior to those of calcium phosphate ceramics. Additionally, nanoscale FT has superior bioactivity in the biological environment to bulk-form FT [34]. Therefore, it is desirable to develop nanoparticle-reinforced PEEK composite with the favorable properties of both PEEK and silicate-based bioceramics (BG and FT). The designed nanocomposites are likely to display enhanced bioactivity and sufficient mechanical strength, which will be of great importance for dental implants applications.

The present study reports the fabrication of either BG- or FT-reinforced PEEK nanocomposites via melt blending followed by compression molding, a common and versatile method easy to scale up. A comprehensive characterization was performed to evaluate in detail the influence of the types and content of nanoparticles on the surface morphology, crystallization behavior, mechanical performance, and bioactive properties in vitro of these biomaterials that are very promising for use in dental implant applications.

## 2. Materials and Methods or Experimental

### 2.1. Materials

Semi-crystalline PEEK fine powder (VICTREX^®^ PEEK polymers, Victrex Technology Centre, Lancashire FY5 4QD, UK) with an average particle size of 50 μm for compression molding was used as the polymer matrix. The density of PEEK is 1.3 g/cm^3^, and its melt viscosity is 350 Pas. Two types of bioactive inorganic nanoparticles, 45S5 bioglass (BG) of chemical composition (45 wt.% SiO_2_, 24.5 wt.% CaO, 24.5 wt.% Na_2_O, and 6 wt.% P_2_O_5_) and forsterite (FT) of chemical structure (Mg_2_SiO_4_) were purchased from NanoTech Egypt Co., Giza, Egypt. The BG nanopowder had an average particle size of 40 ± 4 nm while the average particle size of the FT nanopowder was 30 ± 5 nm. The chemical reagents for the preparation of simulated body fluid (SBF) are sodium chloride (NaCl), hydrochloric acid (HCl), sodium hydrogen carbonate (NaHCO_3_), potassium chloride (KCl), hydrated dipotassium hydrogen phosphate (K_2_HPO_4_·3H_2_O), hydrated magnesium chloride (MgCl_2_·6H_2_O), calcium chloride (CaCl_2_), sodium sulphate (Na_2_SO_4_) and Trisma^®^ base (NH_2_C(CH_2_OH)_3_) ≥ 99.9% and were purchased from Sigma-Aldrich Chemical Co., Darmstadt, Germany.

### 2.2. Fabrication of PEEK/Bioceramics Nanocomposites

The PEEK/BG and PEEK/FT nanocomposites were fabricated using melt blending and compression molding techniques as described in our previously reported data [35]. In the fabrication process, PEEK powder and bioactive nanofillers were dried overnight in a vacuum oven at 120 °C to ensure the complete removal of moisture. Then, they were ball-milled mixed in a planetary ball mill (Emax, Retsch GmbH, Haan, Germany) at 25 °C and 400 rpm for 2 h. The as-milled dried powder was extruded and poured in a split steel mold 10 mm in diameter and 6 mm high via a custom-designed extruder. The powder was compressed at room temperature under a pressure of 35 MPa for 2 min. After cold compaction, the powder was heated to 410 °C while applying a low cavity pressure of about 2 MPa. When the system reached the set temperature, it was held at constant temperature and pressure for 10 min to create homogeneity within the melt. After that, the system was allowed to cool down ordinarily to room temperature under a pressure of 20 MPa. Finally, the mold was opened, and the samples in the form of disks were released and polished with a series of silicon carbide abrasive paper, then rinsed and dried prior to mechanical characterization and in vitro testing (Figure 1). The exact composition of the prepared PEEK/BG and PEEK/FT nanocomposites is presented in Table 1. In this work, the PEEK polymer was designated by a code of the PK. The codes BG and FT were designated to bioglass and forsterite bioactive nanofillers, respectively, followed by a number indicating the wt.% of bioactive nanoparticles. For example, the PEEK/FT nanocomposite with 10 wt.% FT nanoparticles was coded as PKFT-10. In the case of PEEK/BG nanocomposite containing 10% by weight, BG nanofillers were designated as PKBG-10.

### 2.3. Characterization Methods

#### 2.3.1. X-ray Diffraction (XRD)

Crystal structural analysis of the pristine PEEK, as well as PEEK/BG and PEEK/FT nanocomposites, were performed by powder XRD measurements using a diffractometer (XRD-7000, Shimadzu, Tokyo, Japan). The X-ray beam was Cu Kα radiation (λ = 0.1542 nm), operated at 40 kV and 30 mA. The XRD pattern was recorded in the 2θ range from 10° to 50° with a scanning rate of 5° per min. The identification of phases was accomplished by comparing the obtained sample diffraction pattern through standard cards in the International Centre for Diffraction Data database and the Joint Committee on Powder Diffraction Standards (ICDD-JCPDS) database [36].

#### 2.3.2. Scanning Electron Microscopy (SEM)

The surface morphology of micro-roughened PEEK/BG and PEEK/FT nanocomposites was characterized by a scanning electron microscope (SEM) using a JEOL instrument (JSM-5300, Tokyo, Japan), which was run at 25 keV. Prior to SEM observation, all samples were ultrasonically washed for 1 min. and sputter-coated with gold to a thickness of 30 nm.

#### 2.3.3. Surface Roughness Analysis

The surface roughness (Ra) of the samples (*n* = 6 per group) was evaluated (according to ISO 4287-1:1997) by an optical profilometer (MarSurf PS1, Mahr GmbH Göttingen, Göttingen, Germany). Four different locations perpendicular to the surface of each sample were recorded. Mean Ra values were statistically analyzed and used as the final Ra score for each sample [37].

#### 2.3.4. Contact Angle Measurement

Water contact angle experiments in a goniometer digital (RAMÉ-hart Model 190-F2, Succasunna, NJ, USA) were used to determine the surface hydrophilicity of the samples. The static sessile drop technique was performed using the video contact angle method. Six samples in each stage were used to provide an average and standard deviation [33].

#### 2.3.5. Microhardness Measurement

The microhardness of the samples was measured as a Vickers’ hardness number (VHN) (Wolpert micro-Vickers tester, Wolpert Wilson Instruments, division of Instron Deutschland GmbH, Aachen, Deutschland). The indentations were made using a diamond pyramid micro-indenter with a 136° angle between the opposing faces under a load of 200 N applied for a dwell time of 10 s.

#### 2.3.6. Mechanical Tests

Compression analysis was carried out by using a universal testing machine (AG-IS 100KN, Shimadzu Corporation, Kyoto, Kyoto Prefecture, Japan) according to ISO 604: 2002 (Plastics—Determination of compressive properties). The test was performed with a cylindrical sample of dimensions (10 mm diameter × 6 mm height) at a crosshead speed of 1.0 mm/min until the specimen failed. Flexural properties were measured using a three-point bending test method in the same universal testing machine according to ISO 178: 2010 (Plastics—Determination of flexural properties). The test was carried out with a rectangular bar sample of dimension (80 mm × 6 mm × 6 mm) at a crosshead speed of 1.0 mm/min at room temperature. A total of six samples were tested for each material group (PEEK, PKFT, or PKBG) to obtain the average value [24].

#### 2.3.7. Bioactivity Testing

The pristine PEEK, PEEK/BG, and PEEK/FT samples were immersed in a simulated body fluid (SBF), almost equal to that of human blood plasma at 37 °C by water bath to examine the bioactivity. After 7, 14 and 28 days, the specimens were removed from the given solution, gently rinsed with distilled water, and quickly dried and kept in a drying oven at room temperature. The chemical composition of surface deposits was characterized by energy-dispersive X-ray (EDX) microanalysis attached to the SEM. Analysis was performed on uncoated samples at 15 kV for 60 s and the Ca/P ratio on the surface of the samples was determined. After sputter-coated with gold, the microstructures of specimens were examined by SEM equipped with EDX [33].

### 2.4. Statistical Analysis

Statistical Package for the Social Sciences (SPSS) software version 20 (IBM Corp, Armonk, NY, USA) was used to analyze the data. The Kolmogorov–Smirnov (K–S) test was applied to assess the assumptions of normal data distributions and homogeneity of variances. Multiple comparison analyses were performed using a one-way analysis of variance (ANOVA) analysis with *p* = 0.05, Tukey’s post hoc test, and Student’s *t*-test.

## 3. Results and Discussion

### 3.1. Fabrication of PEEK Nanocomposites

PEEK/BG and PEEK/FT nanocomposites were successfully fabricated by the dry mixing of PEEK and bioactive nanofillers using high energy ball milling, followed by melt blending and compression molding. Indeed, the uniform dispersion of the nanofillers particles within the PEEK matrix is crucial for improving the mechanical properties. However, this is laborious in the case of PEEK nanocomposites, as they cannot be processed by solution methods due to the insolubility of PEEK in most organic solvents except concentrated hydrofluoric and methyl sulfonic acids. Hence, the ball milling technique was used to reduce the particle size of pristine PEEK from the millimeter to micrometer scale (~5 μm) and disperse nanofillers particles into the PEEK matrix [35]. Moreover, it is more feasible to fabricate PEEK nanocomposites via the melt blending and compression molding technique. In this work, the content of the nanofillers, either BG or FT, varied at 10%, 20% and 30%. These values were selected because they represent an optimum performance of the uniform dispersion of the nanofillers within the PEEK matrix, which is significant for improving the mechanical properties. However, loaded content exceeding the 30% threshold was discounted to avoid the strong agglomerating tendency of the nanofillers particles [33].

### 3.2. Structural Analysis

Figure 1 shows the XRD spectrum of the pristine PEEK as well as PEEK nanocomposites, PEEK/BG and PEEK/FT. The diffraction peaks, at approximately 2*θ* = 18.5°, 21°, 22.5° and 28.6°, corresponded to the characteristic peaks of PEEK [27]. The XRD peaks of BG nanoparticles were dispersive and there was no sharp diffraction peak, which demonstrated that the BG particles were typical of an amorphous structure (Figure 1a). The diagram confirms the vitreous character of the specimen, as no crystalline peaks occur. At 2*θ* of around 29.24° and 33.75°, there appeared peaks that were typical amorphous structure characteristic peaks of silicate glass, according to ICDD [38]. The results indicated that the PEEK nanocomposite was composed of BG and PEEK.

The crystalline phases of the as-milled PEEK and PEEK/FT were also examined, as presented in Figure 1b. As expected, the pure PEEK exhibited only peaks associated with the semi-crystalline PEEK phase. After the incorporation of FT nanofillers, the PEEK showed additional peaks. It displayed the existence of the FT phase with the most intense peak at 2*θ* = 42.80°, suggesting that the high crystalline FT phase was formed within the PEEK/FT nanocomposites [39].

The PEEK/FT nanocomposites showed higher crystallinity in comparison with the PEEK/BG nanocomposites and pure PEEK. The reason behind this result is that the introduction of FT nanofillers to the PEEK matrix can provide additional nucleation sites within the polymer and thereby increase the overall crystallization of PEEK/FT nanocomposites [20].

However, the addition of BG nanofillers into the polymer matrix decreases the overall crystallinity of the PEEK/BG nanocomposites in comparison to the pure PEEK, which could be attributed to the amorphous nature of the BG nanofillers, which is not expected to show any changes in the crystallinity of pure PEEK. It was reported that the amorphous nanoparticles could, more or less, hinder the mobility of the polymer chain segments during crystallization, especially when the particle concentration is high enough [40,41,42].

### 3.3. Morphological Observation

SEM micrographs of the pure PEEK and PEEK nanocomposites loaded with different bioactive nanofillers contents (10, 20, and 30 wt.%) were captured to investigate the exact microstructure, as shown in Figure 2. The surface of the pure PEEK sample appears dense, compact and smooth as no particles were observed on the surface, as shown in Figure 2a. Generally, the surface of PKBG samples appeared rough, which could be attributed to the lack of good dispersion of BG nanofillers, and consequently, many BG agglomerates and voids were observed in the PEEK matrix. As revealed in Figure 2b,c, the SEM images of the PKBG-10 and PKBG-20 samples showed dispersed particles with high electron density (white shapes) distributed on the nanocomposite surface. However, the SEM image of PKBG-30 showed crystallites with a rosette-like shape, which is attributed to the aggregation of the BG nanofillers causing flake-like larger agglomerates (Figure 2d) [43].

Regarding the SEM images of PEEK/FT (PKFT group), the surface appeared more compact and smoother in comparison to that of the PKBG group, and white particles appear randomly distributed on the surface of the PEEK matrix with no agglomerates shown on the surface, as illustrated in Figure 2e–g.

### 3.4. Surface Roughness (Ra)

Average values of surface roughness for the pure PEEK and PEEK nanocomposites loaded with different BG and FT nanofillers contents are summarized in Table 2. In general, the incorporation of BG or FT nanofillers into the PEEK matrix increased the surface roughness of the nanocomposites in comparison to pure PEEK, which is in good agreement with the SEM micrographs. More specifically, the addition of 10%, 20% and 30 wt.% BG increased significantly (*p* < 0.05) the surface roughness of the PKBG group by 31%, 43%, and 56%, respectively. However, the increase in the surface roughness of the PKFT group was less than that of the PKBG group. It can be noticed that the interactions of BG nanofillers with PEEK are not strong enough to enable good dispersion within the polymer matrix. Furthermore, FT nanofillers are relatively more compatible with the PEEK matrix. These results are in agreement with previously reported data [44,45,46].

It is worth mentioning that surface roughness is thought to be beneficial for the bioactivity of dental implants. Several authors stated, that adding surface roughness will enhance the wettability or hydrophilicity of PEEK dental implants, which in turn, exerts a positive influence to overcome the limited bioactivity of PEEK [47,48]. For example, the surface roughness of PEEK nanocomposites up to 1.96 µm can induce osteogenic differentiation, for maximum bone-like tissue formation [49].

### 3.5. Contact Angle Measurement

The bioperformance of dental implants is affected by surface hydrophilicity, as it promotes the interaction between the dental implant material and the surrounding tissue [48]. Thus, it is important to hydrophilically modify the PEEK implant surface to enhance the biofunctionality. In order to assess the relative hydrophilic/hydrophobic properties of the PEEK/BG and PEEK/FT nanocomposites, the water contact angle of the surface of the nanocomposite was determined. Table 2 represents the variation of the contact angle of various PEEK nanocomposites loaded with different contents of the hydrophilic nanofillers BG and FT. Obviously, the highest contact angle, 94.2°, corresponds to the hydrophobic pure PEEK. The addition of 10, 20 and 30 wt.% BG or FT into the PEEK matrix significantly decreases (*p* < 0.05) the contact angle in comparison to pure PEEK. For example, loading 20 wt.% BG and FT reduced the water contact angle to 66.7° and 70.4°, respectively, revealing the increased hydrophilicity by the addition of nanofillers. It is remarkable that the contact angle of PEEK/BG nanocomposites is significantly less than that of PEEK/FT nanocomposites. The reason behind this result is the relatively more hydrophilic nature of BG in comparison to FT [50].

### 3.6. Microhardness

The surface microhardness assessment could be considered a crucial property when testing and comparing different dental implant materials. This is mainly owing to the critical demand for different dental implants for variable surface treatments to achieve adequate osseointegration and to gain long-term success [51]. The results of the microhardness test for the pure PEEK and PEEK nanocomposites are illustrated in Figure 3. The microhardness values decreased significantly (*p* < 0.05) when BG nanofiller was incorporated into the nanocomposite formulation. For example, the addition of 20 wt.% BG decreased the microhardness value from 35 to 26 Kg/mm^2^, that is, 26% lower than pure PEEK. No significant difference was observed within the PKBT group with increasing the amount of BG. The remarkable decrease in hardness could be explained by the widening of the silicate network reducing the glass density and thus, affecting the mechanical properties of the glass, especially the microhardness. Moreover, the intrinsic brittleness of the BG can negatively affect microhardness [52,53]. On the other hand, the microhardness was increased significantly (*p* < 0.05) by 23% with the incorporation of 20 wt.% FT nanofillers into the PEEK matrix but did not increase further when the FT content was increased to 30 wt.%. This may be due to the higher stiffness of the FT nanofillers. Moreover, the relatively uniform distribution of the FT nanoparticles, as seen in the SEM image, and the decrease in the interparticle distance with increasing the particle loading in the matrix results in an increase in the resistance to indentation of the PEEK matrix.

### 3.7. Mechanical Properties

The compression elastic modulus of pure PEEK and its nanocomposites, fabricated by loading different contents of BG and FT nanofillers, are shown in Figure 4. Particularly, the addition of 20 wt.% BG and FT showed a statistically significant increase in the compression elastic modulus by 25% and 60%, respectively, compared to pure PEEK. However, further increases in both nanofillers’ content (30 wt.%) showed an insignificant decrease in the compression modulus of elasticity. When comparing the elastic modulus of both PEEK nanocomposites groups, it was observed that the elastic modulus of the PKFT-20 sample is significantly higher than PKBG-20.

The flexural strength of pure PEEK and its nanocomposites exhibited similar trends, as displayed in Figure 5. The addition of 20 wt.% BG and FT showed a statistically significant increase in the flexural strength by 5% and 40%, respectively, compared to pure PEEK. However, there was a significant decrease in flexural strength for nanocomposites loaded with 30 wt.% BG and FT nanofillers.

From these results, we can conclude that the PKFT-20 sample has the highest flexural compression elastic modulus and strength among all groups, which could be explained by the more uniform and homogenous dispersion of the FT nanofillers when compared to BG. In addition, the intrinsic brittleness of the BG fillers can negatively affect the mechanical properties [52,53,54,55]. Furthermore, higher nanofillers content (30 wt.%) showed an adverse effect on the mechanical properties, due to the agglomeration of nanoparticles and stress concentration occurring at the agglomeration place [54]. This comes into agreement with authors who reported that lower levels of loading are generally favored using nanoscale particulates, as the filler agglomeration that occurs at high loading levels reduces mechanical properties. Such findings indicated that PEEK nanocomposite containing 20 wt.% FT nanofillers is the optimal ratio where the mechanical properties are similar to the human bone, where the elastic modulus is (6–18 GPa) and the flexural strength is 186–262 MPa [55,56]. Based on the results, the PKBG-20 and PKFT-20 nanocomposites were chosen for the in vitro bioactivity test.

### 3.8. Bioactivity Testing

Figure 6 shows the SEM images of the pure PEEK, PEEK/BG 20 wt.%, and PEEK/FT 20 wt.%, nanocomposite surfaces after immersion in SBF for 7, 14, and 28 days. As revealed in Figure 6(a1–a3), there was no apparent observation of Ca–P deposition on the surface of pure PEEK, even after 28 days of immersion in SBF. In contrast, the aggregation of the needle-like apatite layer started to emerge on the surface of PKBG-20 nanocomposite after 7 days of soaking, as displayed by the SEM micrograph in Figure 6(b1). When the immersion time was extended to 14 days, cauliflower-like clusters were formed on the surface of the PKBG-20 (Figure 6(b2)). After 28 days, a mass of apatite islands formed and covered most areas of the PKBG-20 surface, as shown in Figure 6(b3).

Regarding PKFT-20 nanocomposite, few crystals of apatite islands formed on the surface on day 7, as seen in Figure 6(c1). After 14 days of immersion, the surface topography showed a porous apatite-like layer formed on the surface (Figure 6(c2)). Figure 6(c3) shows clusters of agglomerated particles and an apatite-like layer on the surface after 28 days of soaking in SBF.

The EDX results acquired from the immersed specimens revealed that the formed particle contains Ca and P and their ratio was displayed in Table 3. After 14 days of immersion in SBF, the stoichiometric Ca/P ratio was 1.65 for the PKBG-20 sample. However, after 28 days of soaking, the ratio was 1.67 for PKFT-20, which was similar to the stoichiometric value (1.67) of HA in bone minerals. Furthermore, it is proven that by incorporating the FT or BG into PEEK, the bone-like apatite formation is promptly observed in SBF and it increases with the increase in incubation time. It is worth mentioning, however, that the highly crystalline structure of FT showed less bioactive effects than amorphous BG, which can release phosphate or calcium [57]. Generally, the formation process of HA crystals on the surface of PEEK nanocomposites is affected by two factors: nucleation of HA and the diffusion of Ca and P ions from the inner towards the surface of the nanocomposites [58].

The trend of the increase in apatite formation on the surface of the PKBG-20 sample is due to the presence of Ca and P ions within the composition of the BG, which increases the nucleation rate of HA in SBF. This finding agrees with a previous study, which concluded that the composition features of BG are responsible for making bioglass highly reactive to an aqueous medium [59,60]. On the other hand, for the PKFT-20 sample, the trend is due to the slow dissolution of Mg from the crystalline FT in SBF accompanied by a simultaneous uptake of Ca and P from the solution onto the PEEK nanocomposite surface at the early stages of soaking.

## 4. Conclusions

In this study, by evaluating the microhardness, elastic modulus, and flexural strength of PEEK/BG and PEEK/FT nanocomposites with different BG and FT contents (0–30 wt.%), the 20 wt.% FT was considered to be an optimal content. Compared with pure PEEK, both PEEK/BG and PEEK/FT nanocomposites induced apatite formation after immersion in SBF. More importantly, the rough PEEK/FT nanocomposite loaded with 20 wt.% FT exhibits improved bioactivity by immersing the nanocomposite disks in SBF for 28 days due to the co-effects of rough structure and nano-FT crystals. These results have paved the way for the PEEK-based nanocomposite to be used as a potential substitute for metal implant material in orthopedic and dental applications. However, experiments on the in vivo bioactivity and osteogenic differentiation capacity have to be conducted in order to confirm the bone formation around the PEEK-based nanocomposite implants.

## Figures and Tables

**Figure 1 polymers-14-01632-f001:**
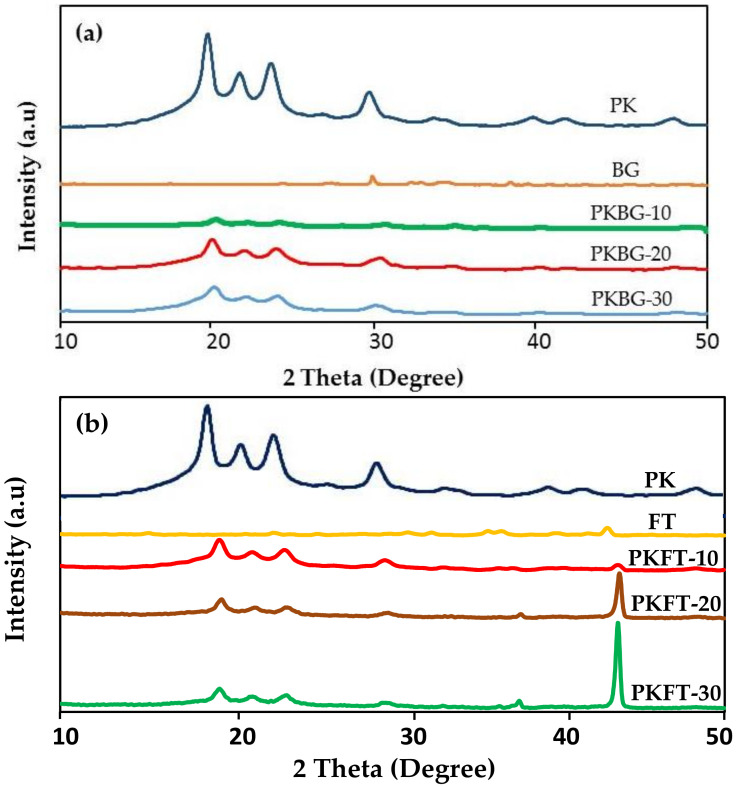
XRD patterns showing the effect of adding BG and FT nanoparticles on the crystallinity of PEEK: (**a**) nanocomposites filled with BG nanoparticles and (**b**) nanocomposites filled with FT nanoparticles.

**Figure 2 polymers-14-01632-f002:**
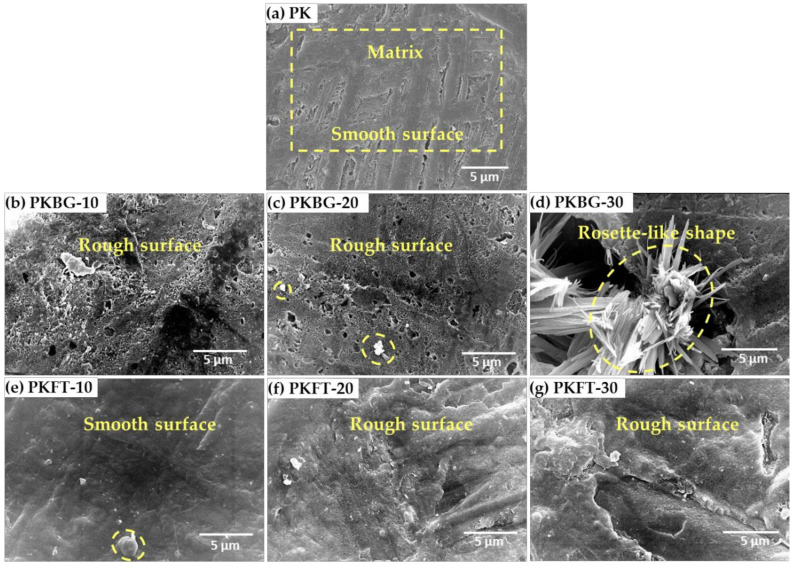
SEM micrographs of the surface morphologies showing details of smooth and rough regions for pure PEEK and PEEK nanocomposites loaded with different content (10, 20, 30 wt.%) of BG and FT nanofillers: (**a**) smooth PEEK, (**b**,**c**) rough PKBG-10 and PKBG-20, (**d**) rough agglomeration of BG nanoparticles on PKBG-30, (**e**) smooth PKFT-10, and (**f**,**g**) low rough PKFT-20 and PKFT-30 with randomly distributed FT nanoparticles within the PEEK matrix.

**Figure 3 polymers-14-01632-f003:**
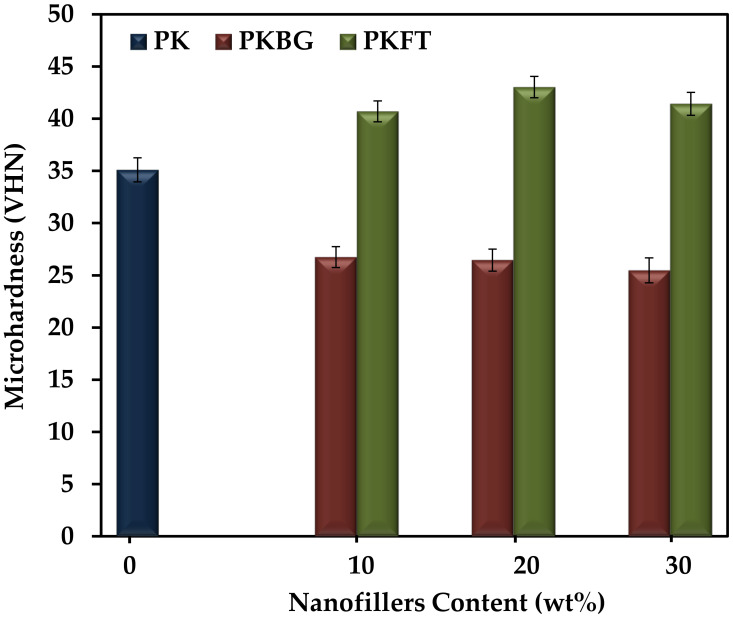
Variations of microhardness (mean ± SD) of the PEEK nanocomposites as a function of the nanofillers content. (*n* = 6, *p* < 0.05).

**Figure 4 polymers-14-01632-f004:**
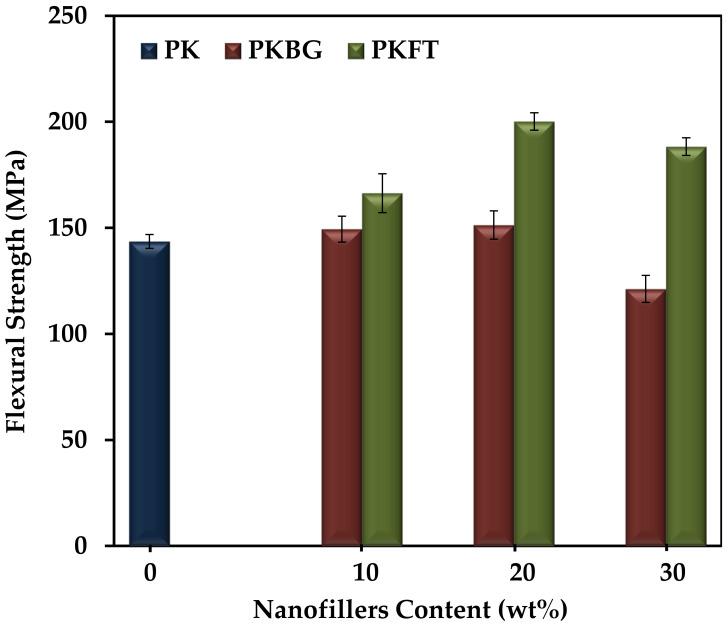
Variations of elastic compression elastic modulus (mean ± SD) of the PEEK-based nanocomposites as a function of BG and FT content. (*n* = 6, *p* < 0.05).

**Figure 5 polymers-14-01632-f005:**
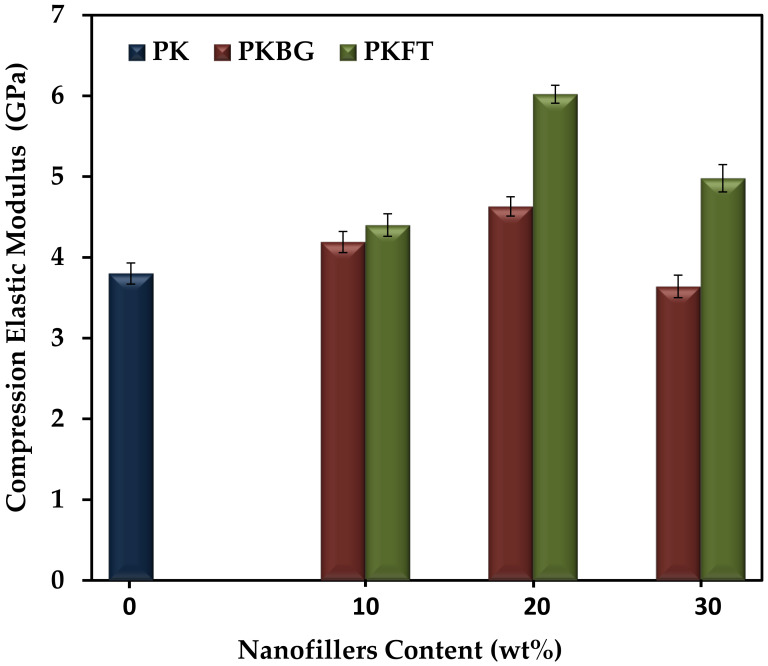
Variations of flexural strength (mean ± SD) of the PEEK/nanocomposites as a function of the nanofillers content. (*n* = 6, *p* < 0.05).

**Figure 6 polymers-14-01632-f006:**
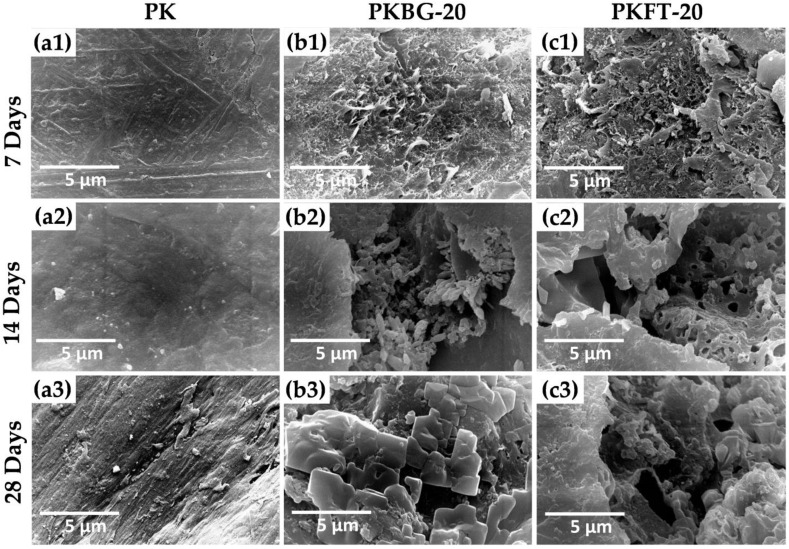
SEM micrographs showing the apatite-formation ability of the pure PEEK and PEEK nanocomposites after immersion in SBF for 7, 14, and 28 days: (**a1**–**a3**) pure PEEK, (**b1**–**b3**) PKBG-20 and (**c1**–**c3**) PKFT-20. No changes were observed on the surface of pure PEEK (**a1**–**a3**) but the PEEK nanocomposites (**b1**–**b3**,**c1–c3**) induced apatite formation after immersion in SBF for all time intervals.

**Table 1 polymers-14-01632-t001:** PEEK/bioglass and PEEK/forsterite nanocomposites formulations (percentage by weight).

		Matrix	Bioactive Filer Content	
Code	Sample	PEEK (wt.%)	BG (wt.%)	FT (wt.%)
PK	Unfilled PEEK	100	0	0
PKBG-10	PEEK/BG 10 wt.%	90	10	0
PKBG-20	PEEK/BG 20 wt.%	80	20	0
PKBG-30	PEEK/BG 30 wt.%	70	30	0
PKFT-10	PEEK/FT 10 wt.%	90	0	10
PKFT-20	PEEK/FT 20 wt.%	80	0	20
PKFT-30	PEEK/FT 30 wt.%	70	0	30

**Table 2 polymers-14-01632-t002:** Mean values and standard deviation n (mean ± SD) for surface roughness and contact angle data of pure PEEK together with various PEEK nanocomposites. (*n* = 6, *p* < 0.05).

Code	Sample	Surface Roughness (Ra) (μm)	Contact Angle (◦)
PK	Unfilled PEEK	1.45 ± 0.17	94.2 ± 1.62
PKBG-10	PEEK/BG 10 wt.%	3.23 ± 0.13	73.1 ± 0.91
PKBG-20	PEEK/BG 20 wt.%	3.51 ± 0.22	66.7 ± 0.70
PKBG-30	PEEK/BG 30 wt.%	3.82 ± 0.34	52.3 ± 1.42
PKFT-10	PEEK/FT 10 wt.%	2.47 ± 0.81	85.1 ± 2.35
PKFT-20	PEEK/FT 20 wt.%	2.61 ± 0.65	70.4 ± 0.32
PKFT-30	PEEK/FT 30 wt.%	2.89 ± 0.14	65.8 ± 0.06

**Table 3 polymers-14-01632-t003:** Ca/P ratio (mean ± SD) on the surface of pure PEEK and PEEK-based nanocomposites, as determined by EDX analysis.

Stoichiometric Ca/P Ratio			
Time in SBF	PEEK	PKBG-20	PKFT-20
7 days	─	1.31 ± 0.08	1.12 ± 0.11
14 days	─	1.65 ± 0.14	1.61 ± 0.17
28 days	─	1.61 ± 0.05	1.67 ± 0.03

## Data Availability

Not applicable.

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
