# Peer review of "Improved Mechanical Properties and Bioactivity of Silicate Based Bioceramics Reinforced Poly(ether-ether-ketone) Nanocomposites for Prosthetic Dental Implantology"

_polymers, 2022, doi:10.3390/polym14081632_

Round 1

Reviewer 1 Report

The manuscript was investigated the mechanical properties and bioactivity of PEEK composite for dental materials. However, please further explain the novelty of this study. More importantly, the biocompatibility test should be performed and included, at least performing a cytotoxicity test. Otherwise, the results are not easy to convince readers. Also, there are a few comments as follows:

  • The abstract should be concise.
  • In the introduction, please add more information for the background. In line 53, the citation is wrong.

  • Regarding surface roughness, please provide ISO standards. How many samples have been measured? Also, please provide the data of Sa, Sdr, Sq, Sv.

Reviewer 2 Report

Poly(ether-ether-ketone) for prosthetic dental Implantology: Enhancement of the mechanical properties and bioactivity. The title doesn’t match the objective. Please edit the title and present in a better way. The authors can add “Poly(ether-ether-ketone) nanocomposite” or similar.

Abstract

The objective is not too clear. Please modify.

Line 30. Is it surface morphology?

Line 36. “It was found that the mechanical properties of the 36 PEEK/forsterite nanocomposites are higher than that of the PEEK/bioglass nanocomposites due to 37 the uniform dispersion of forsterite nanofillers within the polymer matrix.” It is better to use “better mechanical properties” than higher mechanical properties.

Introduction

The Introduction is too short.

It is better to add more recent literatures on the various properties of PEEK nanocomposites and bioactive glass.

https://pubmed.ncbi.nlm.nih.gov/34771318/

https://pubmed.ncbi.nlm.nih.gov/34503046/

https://pubmed.ncbi.nlm.nih.gov/34856483/

https://pubmed.ncbi.nlm.nih.gov/30744178/

https://pubmed.ncbi.nlm.nih.gov/31783484/

https://doi.org/10.1563/aaid-joi-D-16-00072

https://www.sciencedirect.com/science/article/pii/S0264127517305269

Please talk about PEKK. Both PEEK and PEKK are family members of PEAK.

https://www.sciencedirect.com/science/article/pii/S2090123220302137

Line 53. [1] (Ortega-Martínez, J. et al, 2020). Is there citation error?

Line 73-75. “To the best of our knowledge, this is the first time this bioactive forsterite nanofillers 73 have been used in combination with PEEK to produce bioactive PEEK nanocomposites 74 for prosthetic dental implant application.”

Modify this sentence and give scientific clarification.

Method

Why they use only 10, 20, 30%? Need to mention. Is there any reference?

Is it possible to add references for each testing?

Which version of SPSS you used? Please add. And type of ANOVA used.

The authors didn’t add the statistical comparison results. The statistical comparison and the results need to be added in the Figures and the Tables.

Please add the limitations and future perspectives.

Conclusion

The conclusion is long. Make brief and to the point.

Round 2

Reviewer 1 Report

The manuscript can be considered for publication.

Reviewer 2 Report

Many thanks for the revision and incorporating changes